# Electronic data review, client reminders, and expanded clinic hours for improving cervical cancer screening rates after the COVID-19 pandemic shutdowns: A multicomponent quality improvement program

Sue Ghosh[1]*, Jackie Fantes[1], Karin Leschly[1], Julio Mazul[1], Rebecca B Perkins[2]

[1]East Boston Neighborhood Health Center, East Boston, United States; [2]Boston University, Boston, United States

## Abstract

**Background:** To improve cervical cancer screening (CCS) rates, the East Boston Neighborhood Health Center implemented a quality improvement initiative from March to August 2021.
**Methods:** Staff training was provided. A 21-provider team validated overdue CCS indicated by electronic medical record data. To improve screening, CCS-only sessions were created during regular clinic hours (n = 5) and weekends/evenings (n = 8). Patients were surveyed on their experience.
**Results:** A total of 6126 charts were reviewed. Of the list of overdue patients, outreach was performed on 1375 patients to schedule the 13 sessions. A total of 459 (33%) patients completed screening, 622 (45%) could not be reached, and 203 (15%) canceled or missed appointments. The proportion of total active patients who were up to date with CCS increased from 68% in March to 73% in August 2021. Survey results indicated high patient satisfaction, and only 42% of patients would have scheduled CCS without outreach.
**Conclusions:** The creation of a validated patient chart list and extra clinical sessions devoted entirely to CCS improved up-to-date CCS rates. However, high rates of unsuccessful outreach and cancellations limited sustainability. This information can be used by other community health centers to optimize clinical workflows for CCS.
**Funding:** All funding was internal from the EBNHC Adult Medicine, Family Medicine, and Women's Health Departments.

*For correspondence: ghoshs@ebnhc.org

## Editor's evaluation

This study addresses a pertinent and important topic related to prolonged delays in cervical cancer screening and the need to maintain routine and timely screening services in a large health maintenance network in Boston. The findings provide a solid roadmap for implementing simple strategies to help patients return to essential health services.

## Introduction

Nationwide closures at the beginning of the COVID-19 pandemic led to decreases in breast, colorectal, and cervical cancer screenings between 86 and 94% compared to 3-year averages (*Mast et al., 2022*). The postponed screenings have led to backlogs that healthcare systems will need to address as operational changes continue with evolving COVID-19 rates. Federally qualified health centers (FQHCs), which primarily serve low-income and minority communities, have been particularly impacted by the pandemic. Low-income and minority communities had disproportionately high cancer incidence and mortality prior to the pandemic (*Du et al., 2011*; and now could have increased disparities due to reduced access to screening because of COVID-19 *Maringe et al., 2020*).

The East Boston Neighborhood Health Center (EBNHC) is the largest FQHC in Massachusetts. It was established in 1970 and has approximately 170,000 patient visits annually. Its catchment area includes 270,000 patients. Over 70% of the patient population is Latinx. During the first year of the COVID-19 pandemic, most non-urgent in-person medical care was postponed, including cervical cancer screening (CCS). A quality improvement (QI) project was initiated at the EBNHC in March 2021 to improve CCS rates. The purpose of the project was to examine the impact of a QI intervention on improving CCS, as well as to evaluate the effectiveness and sustainability of different methods for addressing overdue screening. The multicomponent intervention utilized best practices as recommended by the Center for Disease Control and Prevention's (CDC's) Community Preventive Services Task Force community guide, including interventions to increase community demand (client reminders), interventions to increase community access (extended hours), and interventions to increase provider delivery of screening services (provider assessment and feedback) (*Centers for Disease Control and Prevention, 2019*). In a meta-analysis examining interventions to increase CCS among ethnic minority women, the intervention with the strongest effect was increased access, followed by community education, and individual counseling/letters. Combining multiple strategies did not increase CCS for Hispanic patients (*Han et al., 2011*). Taking this data into consideration, the clinical leadership at the EBNHC critically evaluated the operational and financial feasibility of various interventions during the COVID-19 pandemic and the CCS project was designed as a descriptive study. This article describes the project's background, lessons learned, and new initiatives the EBNHC will use to improve CCS rates in the future.

## Methods

### Setting: Current CCS program at the EBNHC

CCS is performed by a diverse team of healthcare providers at the EBNHC, including physicians, nurse practitioners, certified nurse midwives, and physician assistants. These screenings take place within the Adult Medicine (AM), Family Medicine (FM), and Women's Health departments. Additionally, a smaller number of CCS are conducted in the Pediatrics Department for patients over the age of 21 who still receive care there. CCS can potentially be conducted during all types of visits. To assist in ensuring timely screenings, our electronic medical record (EMR) system, EPIC, utilizes a care gap alert feature. This feature identifies patients who are overdue for CCS based on the health maintenance aspect of their medical records. The 2018 United States Preventative Service Task Force CCS Guidelines are used as a basis for the EPIC care gap frequency. The CCS care gap frequency is manually changed by providers for patients who require increased CCS based on previous abnormal cytology/HPV results/cervical pathology. A CCS care gap alert fires when it is 1 d overdue based on the set frequency and the date of the patient's last CCS.

The Quality Department at the EBNHC closely monitors CCS metrics, specifically the percentage of CCS performed, to ensure high-quality care. At present, there are no other routine workflows associated with CCS.

The data for this project was collected from the EMR reports. The analysis of the data involved descriptive statistics, and no statistical modeling was conducted.

### Preparatory work: Demonstrating project need

An EPIC work-bench report (WBR) was run to identify active EBNHC patients (defined as at least one visit to a primary care department in the past 18 mo) whose health record stated that CCS screening was overdue. This list had over 8000 patients as of January 2021. The provider tasked with examining

the EBNHC's overdue CCS issue reviewed 1600 charts and noted that over 80% were correctly identified as overdue. This data was presented to the EBNHC clinical and administrative leadership in February 2021. A proposal was made to (1) validate the overdue CCS list, (2) create CCS-only clinics, and (3) improve clinical and electronic CCS workflows. This project was approved by the Project Steering Committee March 2021. This article reports efforts on chart validation and CCS-only clinics. A total of 1600 charts were initially reviewed to determine the need for the QI project.

## Data validation: Creation of outreach list

Starting in March 2021, a 21-provider team was assembled to review 6126 charts of patients identified as overdue for CCS using the report generated by EPIC. The team consisted of nurse practitioners, physician assistants, and physicians from AM, FM, and OB/GYN departments. The team was trained on how to review charts to confirm the need for CCS by the project lead during a 1-hr teaching session. One hundred medical record numbers from the overdue CCS list were sent approximately every 2 wk to each provider to review. A new list of patients was sent to the providers once they reviewed the previously given list. This was completed from April 30 to June 29, 2021, until the entirety of the overdue CCS list was reviewed. All providers received overtime pay from their respective departments for the hours used to review these patient records. A messaging pool was created in the EMR for providers to ask gynecologists questions about CCS necessity or frequency during the validation process. Three staff gynecologists volunteered for this task. A validated list was created of patients overdue for CCS.

## Staff training: Increasing awareness among clinicians

The importance of the problem, magnitude of the deficit, and goals of the QI project were presented to all staff during quarterly staff meetings by the Chief Medical Officer.

## Patient services: CCS-only clinics

To address the backlog of patients overdue for CCS, screening-only clinics were created. To maximize accessibility for patients, two different time blocks were used: regular clinic hours and evenings/weekends.

## Regular clinic hours

From March to April 2021, the OB/GYN department organized five CCS-only clinic sessions during regularly scheduled clinical hours. The OB/Gyn staff (medical assistants and front-desk staff) outreached to patients from the validated list. Because these clinics occurred during regular clinic hours, the normal workflow for patient outreach and scheduling was used and no additional staffing or training was required. The patients outreached for these clinics all had a history of abnormal CCS.

## Evenings/weekends

From May to July 2021, eight additional CCS-only clinic sessions were organized outside of regularly scheduled clinical hours (evenings/weekends). To organize and run the evening/weekend CCS-only sessions, an operations team was created. The team consisted of the CCS project lead, the OB/Gyn clinical leader, operations managers from OB/Gyn and FM, and the OB/Gyn clinical supervisor. Weekly meetings were held from April 2 to July 23, 2021. A dedicated outreach team was also created from the OB/Gyn department: an outreach lead (OB/Gyn clinical supervisor), a medical assistant, and two front-desk staff. The outreach team was responsible for calling patients from the validated list to schedule and keep track of outreach outcomes. A smart phrase was created for these outreach calls. Staff also placed CCS clinic reminder calls 1 d before the scheduled CCS-only clinic. If a patient preferred to have her CCS done by her primary care provider, the message created by the new smart phrase was forwarded to the appropriate departmental pool to inform them to call the patient. The IT department created a new resource schedule in the FM department for these clinics. The data management and analysis were done by the outreach and project leads.

The staffing for the evening/weekend CCS-only clinics was organized by the project lead. Two rotating OB/Gyn front-desk staff were used for front-desk staff. A total of 6 rotating medical assistants and 14 staff providers (NPs, PAs, and MDs) from OB/Gyn, FM, and AM departments were used. Three FM MD residents and two FM NP residents also participated. All clinic staff were paid overtime for the

hours spent in the clinic (residents are not clinic staff and did not receive overtime pay). An orientation document regarding the CCS-only clinics was created by the project lead and emailed to participating providers several days before the clinic to review. A smart phrase for a CCS-only EMR note was also created for those providers who wanted to use it.

The evening/weekend sessions were done at different times to evaluate optimal timing and staffing for possible future permanent CCS clinics. All clinics were held in the FM clinic due to many exam rooms. A total of four Wednesday clinics were held from 5:20 to 8:00 pm; each included 48 appointments. Staff for Wednesday even clinics included six providers, four medical assistants, and one front-desk staff. A total of four Saturday morning clinics were held from 8:00 am to 12:00 pm. Staff included five providers, three medical assistants, and one front-desk staff. The first two Saturday clinics had 72 appointments. The last two Saturday clinics were overbooked with 73 and 79 patients, respectively, based on the number of missed appointments.

## Patient satisfaction

The operations team wanted to collect patient satisfaction data on the evening/weekend CCS clinics to improve clinics in the future. The CCS clinics which took place during regular office hours had already been completed, thus no survey data was collected for those sessions. A patient survey was created with leaders of the CCS project operations team, the director of clinical compliance and risk management, and the Crossroads Group survey company. The survey was created in English and Spanish (*Figure 1*). For the first two CCS clinics done in evenings/weekends, May 5 and 19, the survey was texted to the patients who attended the clinic by the Crosswords Group. For the remaining CCS clinics, a paper version of the survey was handed out to patients when they were being roomed. Following their visits, patients placed the completed surveys into a marked box, and the surveys were subsequently sent to the Crossroads Group for data analysis.

The Project Steering and Patient Care Committees at the EBNHC gave ethical approval of this QI project. This study presents a retrospective analysis of a QI intervention. Due to the nature of the QI intervention, obtaining patient consent was not applicable. The retrospective analysis was conducted on deidentified data. Permission to report deidentified aggregate data and operational details was given by the Chief Medical and Chief Quality Officers.

## Results

A project flowchart is shown in *Figure 2*. It is a summary of the initial chart review, the chart validation, and the patient outreach for the CCS clinics.

## Patient demographics

The patient demographics captured for this project included age information. The median age of the patients was 36 y, range 21–64 y. The insurance status for the EBNHC patient population in general was the following: none 20%, Medicaid 36%, Medicare 5%, both Medicaid and Medicare 2%, other public insurance 4%, and private insurance 34%. The race demographics for the EBNHC patient population was the following: Latinx 70%, White 17%, Black/African American 4%, Asian/Pacific Islander 1%, other <1%, and unreported 4%.

## Data validation/reason for missed CCS

Because all active EBNHC patients had at least one visit in primary care the past 18 mo, all had theoretically one or more opportunities for CCS. Therefore, 118 patients with an overdue CCS were randomly selected for a closer review to evaluate why CCS was not done. Among these 118 patients, 20% did not need a CCS: 14% due to incorrect CCS frequency in the healthcare gap; 3% of patients had CCS done at an outside clinic that was not seen in the patient EBNHC chart; 3% of patients were not part of the EBNHC active patient panel; and 1% of patients were >65 years old or were status post a hysterectomy and did not have CCS health gap turned off. Among those who were due for CCS, several reasons were identified for not completing the screening. In 30% of encounters, the overdue CCS was not noted by the provider, while in 16% of encounters, the overdue CCS was noted but not addressed due to other medical concerns. Patient-related issues were noted in the remaining 54%

## Pap Clinic Patient Satisfaction Survey

*Patient Survey - English*
*PAP Clinic Encounters*

Thank you again for completing your last Pap screening with East Boston Neighborhood Health Center. To help us continually improve, please answer a few short questions about your experience. Your responses will be processed by a third-party survey research firm with results combined and reported with those of other respondents.

1: How helpful was it to you that the center reached out to you to offer a Pap test?

❏ Very Helpful ❏ Somewhat Helpful ❏ Not Helpful

2: Do you think that you would have scheduled a Pap test in the next 6 months if the center had not reached out to you to schedule?

❏ Yes ❏ No ❏ Don't Remember/Not Sure

3: When you were contacted by phone, how would you rate the courtesy and helpfulness of the phone attendant who scheduled your test?

❏ Poor ❏ Fair ❏ Good ❏ Excellent ❏ NA

4: How would you rate the convenience of the day and time that you scheduled the test?

❏ Poor ❏ Fair ❏ Good ❏ Excellent ❏ NA

5: About how long did the entire test take from the time you arrived until you left?

❏ Less than 30 min ❏ Over 45 min - 1 hour ❏ Over 2 hours
❏ Over 30 min - 45 min ❏ Over 1 hour - 2 hours ❏ Don't Remember/Not Sure

　　5a: How would you rate your satisfaction with this duration of _____ for your Pap test?

❏ Poor ❏ Fair ❏ Good ❏ Excellent ❏ NA

6: How would you rate the courtesy and helpfulness of the Medical Assistant who helped the provider administer the Pap test?

❏ Poor ❏ Fair ❏ Good ❏ Excellent ❏ NA

7: How would you rate the courtesy and helpfulness of the Provider who administered the Pap test?

❏ Poor ❏ Fair ❏ Good ❏ Excellent ❏ NA

8: How would you rate the overall quality of how the Pap test procedure was administered?

❏ Poor ❏ Fair ❏ Good ❏ Excellent ❏ NA

9: Were all of your expectations met during the Pap test?

❏ Yes ❏ No ❏ Don't Remember/Not Sure

　　[If No] Please describe how the center could have better met your expectations during this visit:

　　_______________________________________________
　　_______________________________________________
　　_______________________________________________

10: How likely will you be to schedule another Pap test with East Boston Neighborhood Health Center in the future when you receive a reminder?

❏ Very Likely ❏ Somewhat Likely ❏ Not Likely ❏ Not Sure (Live out-of-town)

11: How would you rate your overall satisfaction with your Pap test experience?

❏ Poor ❏ Fair ❏ Good ❏ Excellent ❏ NA

12: Do you have any comments or suggestions related to this experience?

_______________________________________________
_______________________________________________
_______________________________________________
_______________________________________________

Would you like for someone from the center to contact you to discuss any concerns expressed in this survey?

❏ Yes ❏ No

**Figure 1.** Patient satifaction survey. Patient survey given to patients who attended the cervical cancer screening (CCS) clinics in the evenings/weekends. The survey was also given in a Spanish version for patients who identified their preferred language of care as 'Spanish' in the electronic medical record (EMR).

The online version of this article includes the following source data for figure 1:

**Source data 1.** Editable version of patient satisfaction survey.

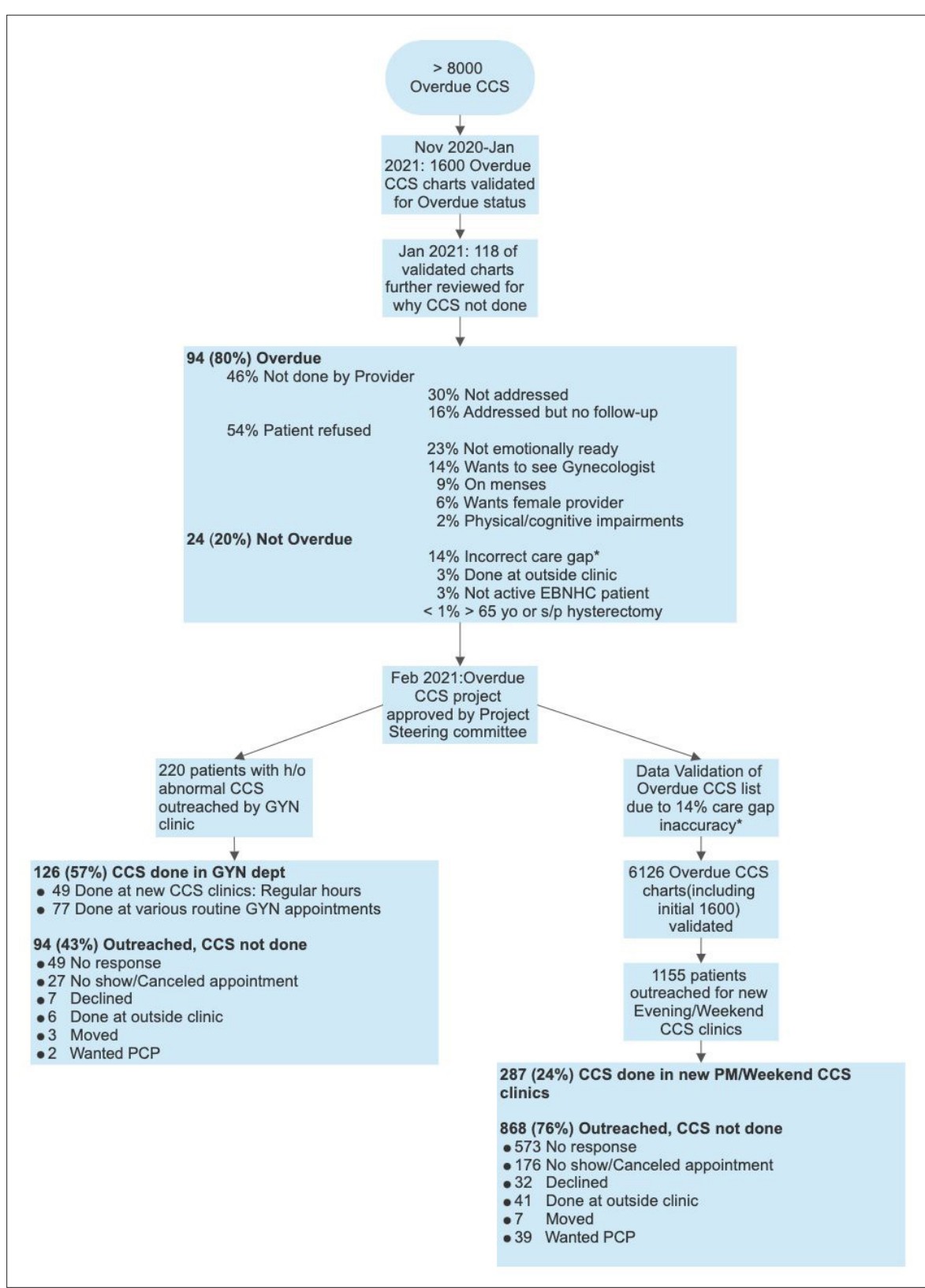

**Figure 2.** Overdue cervical cancer screening (CCS) project flowchart. This flowchart is a summary of the overdue CCS project, including a summary of why CCS was not done for the 118 validated charted reviewed and a summary of the patient outreach for the CCS clinics done in the gynecology clinic (220 patients) vs. those done in the evening/weekend clinics (1155 patients).

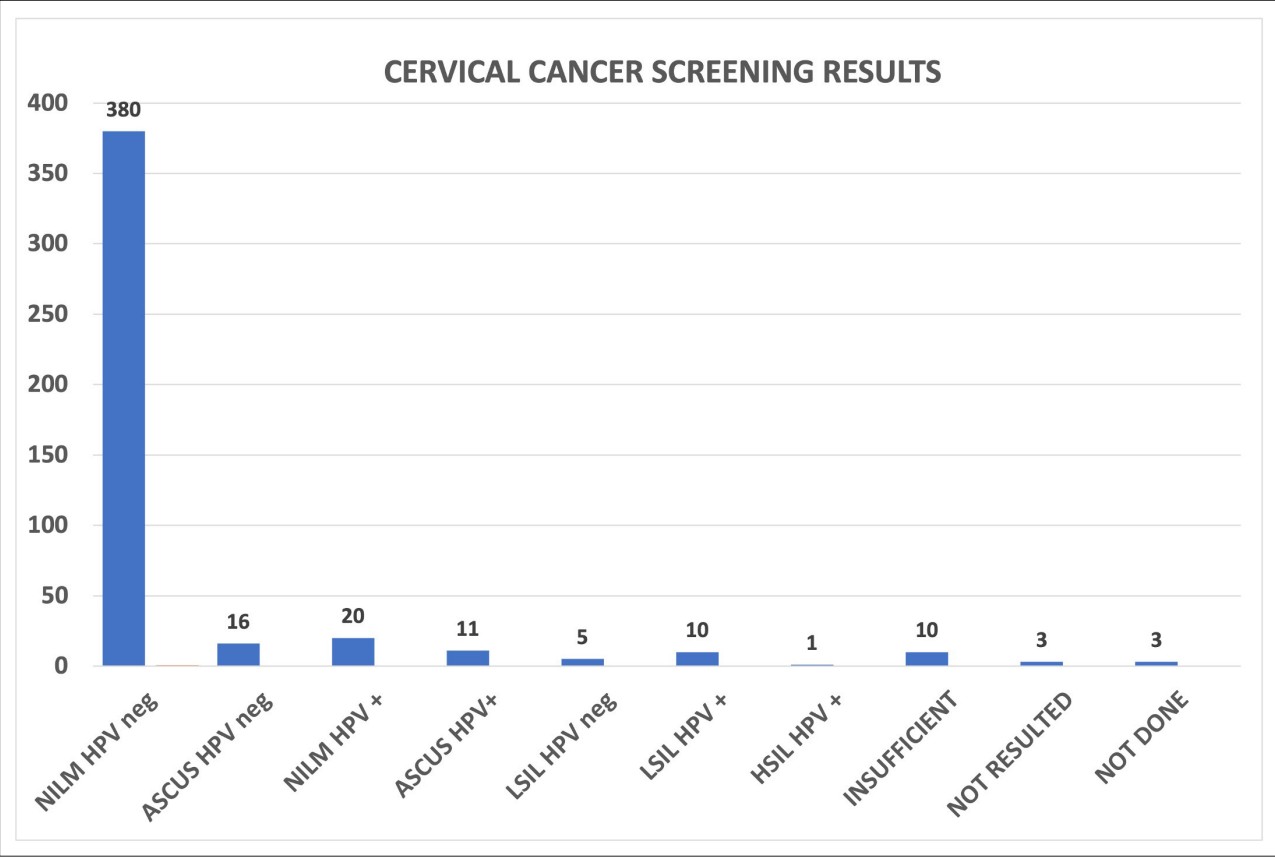

**Figure 3.** Cervical cancer screening results from cervical cancer screening (CCS) clinics. The CCS cytology results from the 459 screenings done during this project.

The online version of this article includes the following source data for figure 3:

**Source data 1.** Spreadsheet with screening results from CCS clinics.

of cases: not emotionally ready (23%), desired gynecologist (14%), on menses (9%), desired female provider (6%), and physical and cognitive impairments (2%).

## CCS results and rates

A total of 459 CCSs were done during this project. This included 126 CCS during regular clinic sessions, 287 done during evening/weekend clinics, and 46 done by the EBNHC PCPs during the project period. The results were as follows: 380 (83%) NILM HPV-neg; 16 (3%) ASCUS HPV-neg; 20 (4%) NILM HPV+; 11 (2%) ASCUS HPV+; 5 (1%) LSIL HPV-neg; 10 (2%) LSIL HPV+; 1 (<1%) HSIL HPV+ HPV 16-neg; 10 (2%) insufficient; 3 (<1%) not resulted; 3 (<1%) ordered but not done; and 1 (<1%) not done (*Figure 3*).

The percentage of all active patients at the EBNHC defined as having had a primary care visit in the past 18 mo who were up to date with cervical cancer screening was 63.5% (36,824 active patients up to date on CCS /57,965 active patients who are eligible for CCS) in October 2020 (nadir of the pandemic) and 68.2% at the start of the project in March 2021 (36,432/53,457). Following the months of the data validation and CCS clinics, the up-to-date screening rate in August 2021 increased to 72.7% (39,040/53,678) (*Figure 4*). To provide a contextual understanding of the CCS rates, we analyzed the annual number of overdue CCSs (*Figure 5*). These numbers represent the number of overdue CCSs by the end of each December. Initially, there was an increase in overdue CCS because of the COVID-19 pandemic. However, following the completion of the overdue CCS project in late summer 2021, there was a subsequent decrease in the number of overdue CCS. Once the project concluded, the level of emphasis on pap screening gradually decreased and reached a point close to the numbers observed prior to the initiation of the project.

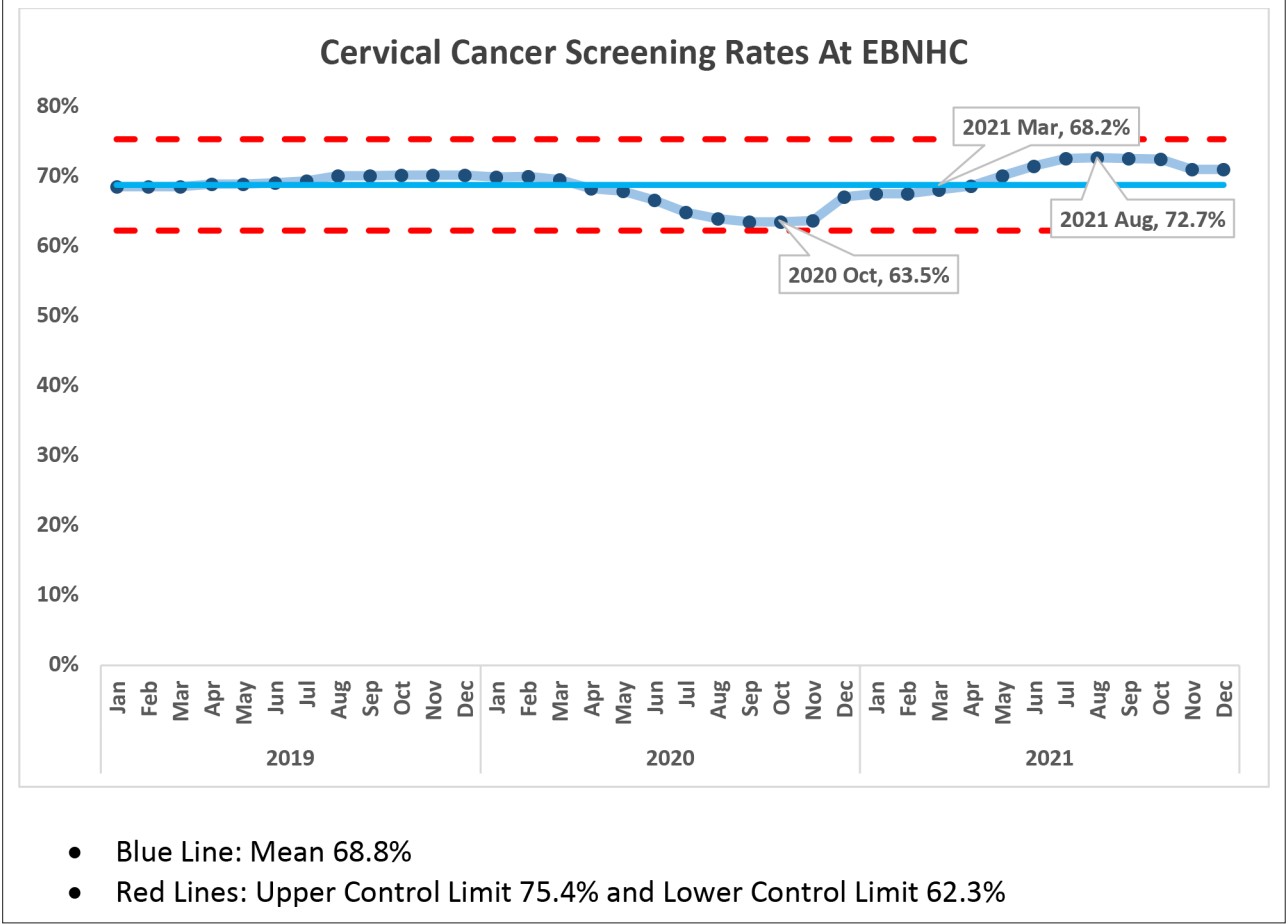

- Blue Line: Mean 68.8%
- Red Lines: Upper Control Limit 75.4% and Lower Control Limit 62.3%

**Figure 4.** Cervical cancer screening (CCS) rates at the East Boston Neighborhood Health Center (EBNHC). The proportion of active patients up to date with CCS each month from 2019 to 2021. The proportion of patients up to date with CCS was the lowest in October 2020 at 63.5%, representing a nadir for the health center during COVID-19. The overdue CCS project started in March 2021 when the CCS up-to-date rate was 68.2%. After completion of the project in August, 2021, the CCS up-to-date rate had increased to 72.7%.

The online version of this article includes the following source data for figure 4:

**Source data 1.** Spreadsheet of CCS rates from 2019 - 2021 for EBNHC.

## CCS-only clinics

The CCS-only clinics were effective in addressing the backlog of patients. However, they involved a significant investment of clinic resources for staffing and outreach, and many patients could not be reached or missed their scheduled appointments.

## Regular clinic hours

The newly created patient outreach team initially called 220 patients all with a history of abnormal CCS. In total, 126 (57%) patients had CCS done. A total of 49 (39%) CCS were done during CCS-only clinics in the Gynecology Department. The remaining 77 (61%) patients were seen by various gynecology providers during their regularly scheduled clinics. Anecdotally, patients and providers felt the CCS clinics were done efficiently. Also, 15% of patients wanted to discuss other gynecological issues during their CCS visit, which was accommodated. Among the 94 patients who did not complete CCS, 22% patients were not reached, and a letter was sent, or a voicemail was left, 12% canceled or missed their appointments, 3% declined screening, 3% had their CCS done at an outside clinic, 1% moved, and <1% of patients wanted to have their CCS done by their PCP (*Figure 6*).

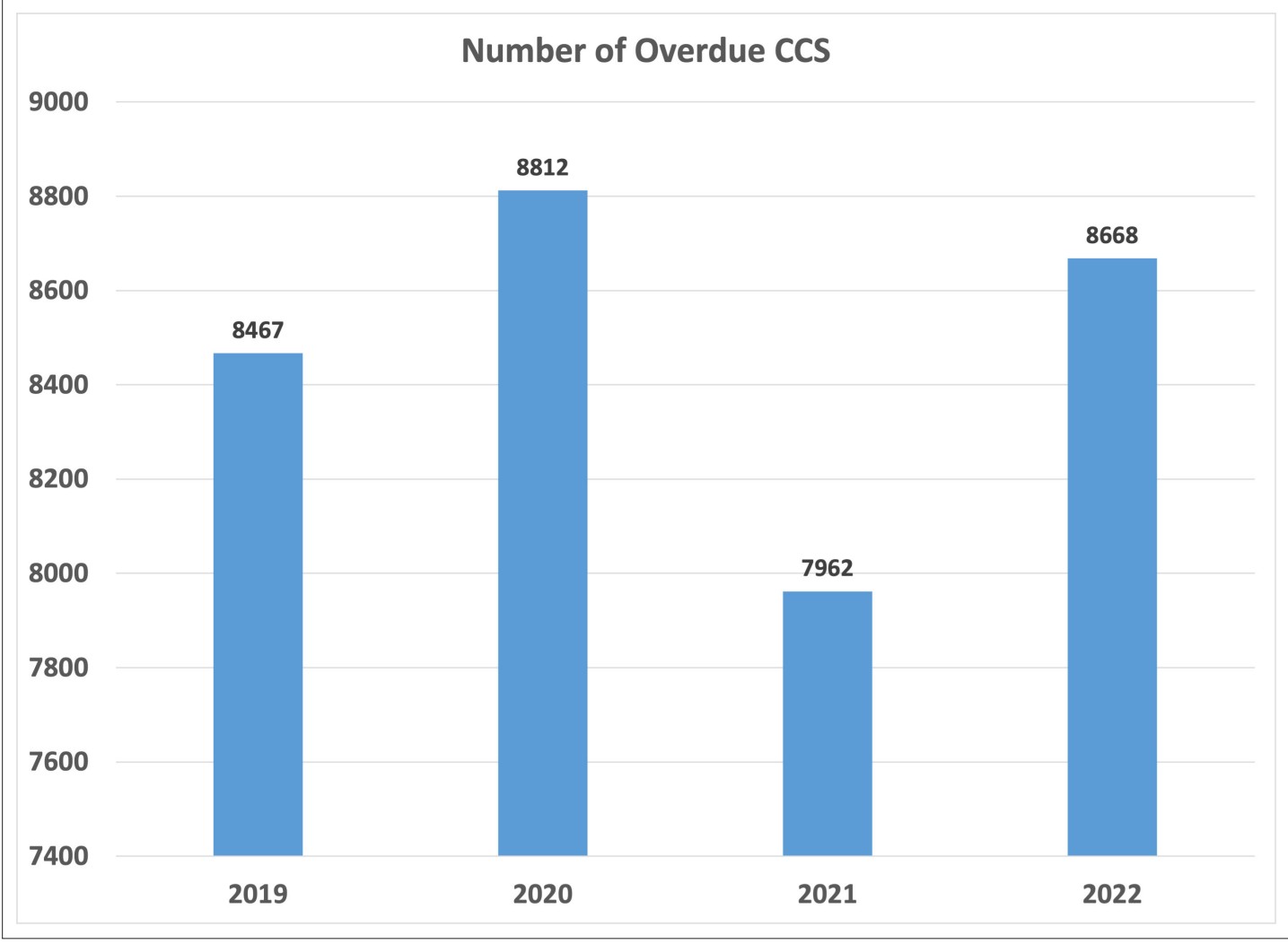

**Figure 5.** Annual overdue cervical cancer screening (CCS) number. Each bar represents the number of patients flagged as overdue for CCS on December 31 per year. From 2019 to 2020, there was an increase in the number of overdue CCS likely due to the COVID-19 pandemic. From March to August 2021, the overdue CCS project was done. The number of overdue CCS decreased by 1309 by December, 2021. Once the project concluded, the level of emphasis on pap screening gradually decreased and reached a point close to the numbers observed prior to the initiation of the project.

The online version of this article includes the following source data for figure 5:

**Source data 1.** Spreadsheet with overdue CCS number for 2019 - 2022 at EBNHC.

## Evenings/weekends

A total of 1155 patients were called from the validated overdue CCS list, which included patients with and without prior abnormal results who were overdue for CCS. These patients were scheduled into one of eight extra clinical sessions. Of these, 24% of patients completed their CCS, 50% were not reached, 15% canceled or did not show up for their appointment, 4% had the CCS done at an outside clinic, 3% wanted their PCP to do their CCS, 3% declined screening, and <1% had moved (*Figure 7*).

Among the 462 patients with scheduled visits, approximately 62% completed screening and 38% canceled or did not show up to their appointments. We examined whether there was a pattern for the missed appointments at each time slot to decide whether modifying the number of patients per time slot would improve attendance. Examining the evening and Saturday morning missed appointments did not show a pattern (*Figures 8–10*).

## Patient satisfaction

During the eight CCS clinics held in the evenings/weekends, a total of 194 surveys were collected from the 287 patients where a CCS was obtained (response rate 68%). For the May 5 and 19 clinics, five surveys were obtained through text messaging for each session. This led to a decision to switch to distributing paper surveys to patients during the clinic for last six evening/weekend sessions. The satisfaction scores were 83% excellent (121/146 surveys) and 17% good (25/146 surveys). Patients largely rated their interactions as excellent with phone attendants (86%, 166/193) and healthcare providers (85%, 162/191), and (77%, 146/190) found the test to be convenient. Nearly all (93%, 168/181) reported that their expectations were met, and 95% stated they were very likely to make another CCS appointment in the future (95%, 134/141). Only 42% (69/166) stated that they would have scheduled a CCS appointment without EBNHC outreach. Note that denominators are different as not every patient who turned in survey answered all the questions of the same questions.

The high scores from the evening/weekend CCS clinics suggest that the clinic was delivering high-quality care, meeting patient needs, and creating a positive patient experience. These scores also indicated that patients are likely to have trust in the clinic and may be more inclined to recommend it to others. We can also use the scores for marketing future CCS clinics.

The extent of effort required to achieve the project's goal of enhancing CCS rates was diverse. A comprehensive overview of effort levels, their corresponding value, and potential future avenues for work can be found in *Table 1*.

**Table 1.** Effort vs. value summary for overdue cervical cancer screening (CCS) project.

|  | Low value | High value |
|---|---|---|
| Low effort |  | Identify the number of overdue CCS for the health center, per department, and per provider pulled from an overdue CCS report in the EMR.<br>Sample overdue CCS list to evaluate why CCS are not being done. Use results to focus areas of improvement.<br>Utilize resident clinics for overdue CCSs.<br>Prioritize patients for outreach (prior abnormal, >5 y overdue, etc.) |
| Medium effort |  | Patient satisfaction surveys<br>CCS clinics during regular hours<br><br>• New providers or providers who are returning from leave<br><br>Utilize EMR patient outreach abilities to automate and centralize outreach for overdue CCSs. |
| High effort | Evening and weekend CCS-only clinics<br><br>• Telephone outreach to >1000 patients<br>• Unable to reach ~50% of patients<br>• Overtime providers pay<br><br>38% average no show rate among scheduled patients | Single designated provider makes list of all patients overdue for CCS and review for accuracy<br><br>• Over 6000 charts reviewed by 21 providers<br>• Increases provider awareness of scope of problem, leading to increased CCSs done on own<br>• Educates providers on how to review chart to find all aspects of cytology/HPV result/pathology needed to confirm overdue CCS<br><br>Data collection, management, and analysis<br><br>• Increase buy-in of clinical and administrative leaders by presenting data<br><br>Cervical cancer navigator<br><br>• Centralize review and outreach of abnormal cytology, HPV, and pathology results<br>• Centralize data collection and analysis<br>• Utilize population health department if present<br><br>Patient education campaign on CCS<br><br>• Grants for community health centers, cancer screening catch-up after pandemic, high-risk populations |

EMR, electronic medical record.

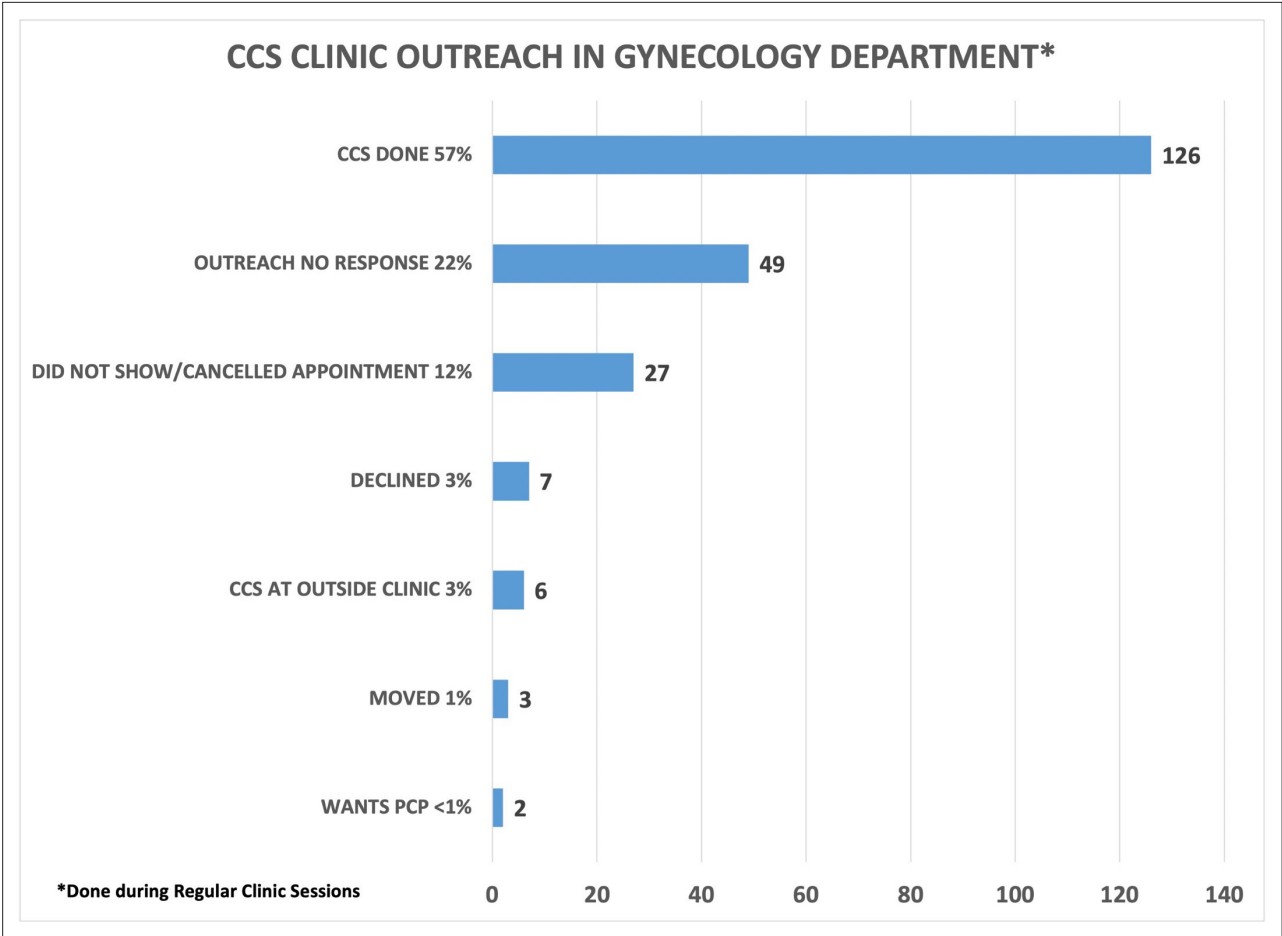

**Figure 6.** Cervical cancer screening (CCS) clinic outreach in the Gynecology Department. The gynecology clinic outreached to 220 patients with a history of abnormal CCS from the validated overdue CCS list to schedule into one of five CCS clinics done during regular gynecology clinic hours. 57% of patients had the CCS done.

The online version of this article includes the following source data for figure 6:

**Source data 1.** Spreadsheet with CCS Outreach for CCS clinics done in the Gynecology Department.

**Source data 2.** Spreadsheet for Outreach for CCS clinics done in Gynecology Department part 2.

**Source data 3.** Spreadsheet of outreach data for CCS clinics done in Gynecology Clinic part 3.

**Source data 4.** Spreadsheet for outreach data for CCS clinics done in Gynecology Department part 4.

**Source data 5.** Spreadsheet for outreach data for CCS clinics done in Gynecology Department part 5.

## Discussion

The overdue CCS project was designed to explore the impact of a QI intervention on the enhancement of CCS. Concurrently, it aimed to assess the effectiveness and long-term viability of different methods for addressing overdue screening. The EBNHC's ability to increase the proportion of our active patient population up to date with CCS by 4.5% during the COVID-19 pandemic was multifactorial. Validating overdue patient lists removed 15% of patients inaccurately flagged as needing screening, which created an accurate denominator to determine up-to-date status. Creating directed outreach and implementing CCS-only sessions led to the completion of 459 CCS. In addition, return of patients to primary care as preventive services reopened in after widespread COVID-19 vaccination in 2021; increased awareness among PCPs of the importance of CCS due to both direct staff outreach and participation in the chart validation; and support from clinical leadership may have led clinicians to capitalize on opportunities to perform CCS when patients presented for clinical care.

The CDC Community Preventive Services Task Force Community Guide recommends multicomponent interventions to increase cancer screenings (***Cancer screening: Multicomponent Interventions***

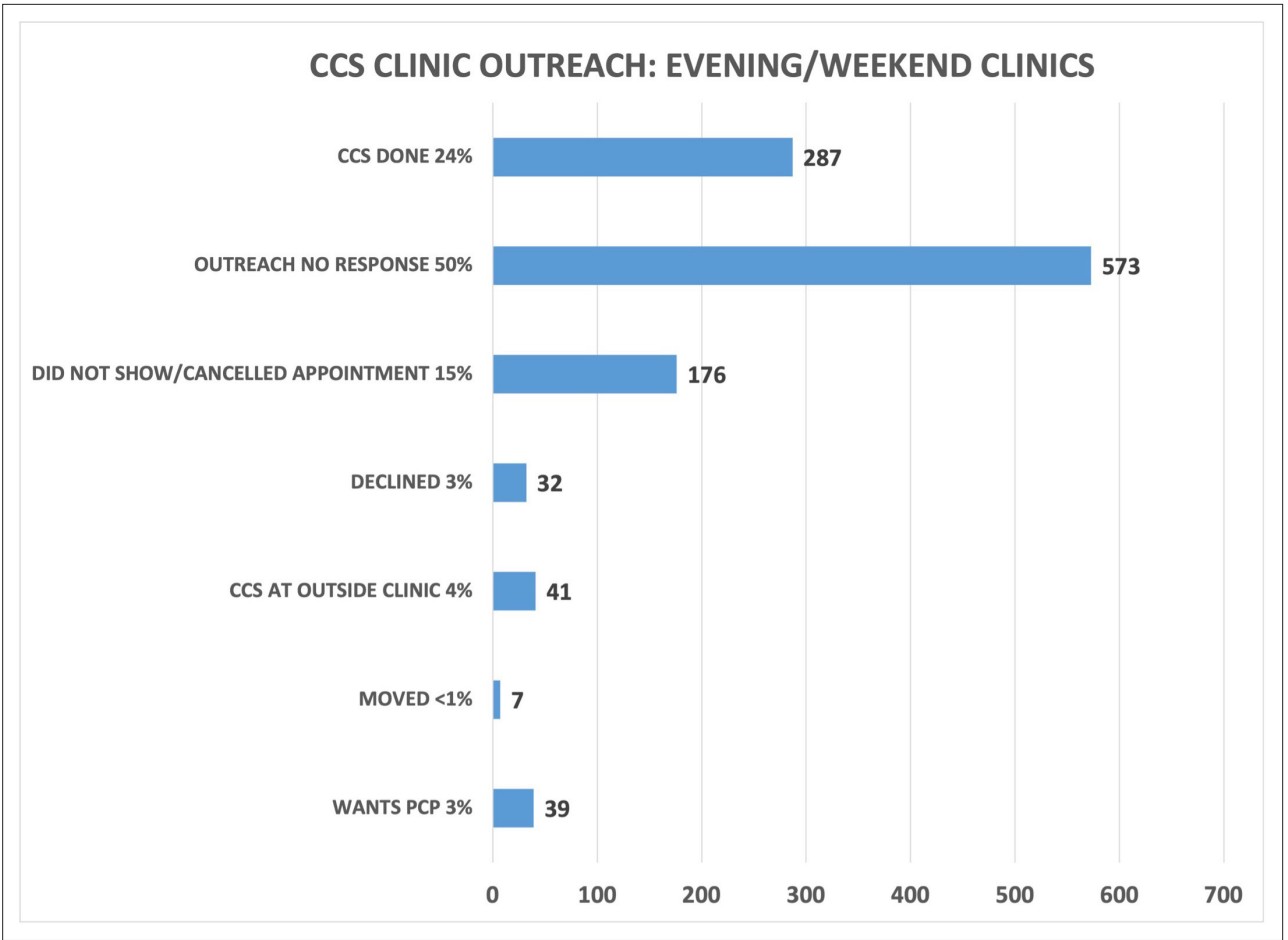

**Figure 7.** Cervical cancer screening (CCS) clinic outreach for evening/weekend sessions. The project outreach team attempted to call 1155 patients from the validated overdue CCS list to schedule into one of eight CCS clinics done during evening/weekend hours. 24% of patients had the CCS done.

The online version of this article includes the following source data for figure 7:

**Source data 1.** Spreadsheet for outreach data for CCS clinics: weekend/evening clinic 1.

**Source data 2.** Spreadsheet for outreach data for CCS clinics: weekend/evening clinic 2.

**Source data 3.** Spreadsheet for outreach data for CCS clinics: weekend/evening clinic 3.

**Source data 4.** Spreadsheet for outreach data for CCS clinics: weekend/evening clinic 4.

**Source data 5.** Spreadsheet for outreach data for CCS clinics: weekend/evening clinic 5.

**Source data 6.** Spreadsheet for outreach data for CCS clinics: weekend/evening clinic 6.

**Source data 7.** Spreadsheet for outreach data for CCS clinics: weekend/evening clinic 7.

**Source data 8.** Spreadsheet for outreach data for CCS clinics: weekend/evening clinic 8.

*Cervical Cancer, 2022* ). The main components of our intervention focused on increasing community demand (client reminders), interventions to increase community access (extended hours), and interventions to increase provider delivery of screening services (provider assessment and feedback). Establishing the intervention had several steps. First, we had to establish buy-in from leadership. Our patient population is >70% Latinx. The Latina population have the highest incidence of new cervical cancer cases and the second (to black women) highest incidence of cervical cancer deaths in the United States (*Centers for Disease Control and Prevention, 2019*). We therefore felt this issue was a priority for our patient population. A single provider's review of the overdue CCS list showed high rates of overdue screenings, which led to the buy-in from the clinical and academic leaders to proceed with the project. Understanding the scope of the issue and catching up with the CCS rate were important to prevent an increase in cervical precancer and cancer in our community.

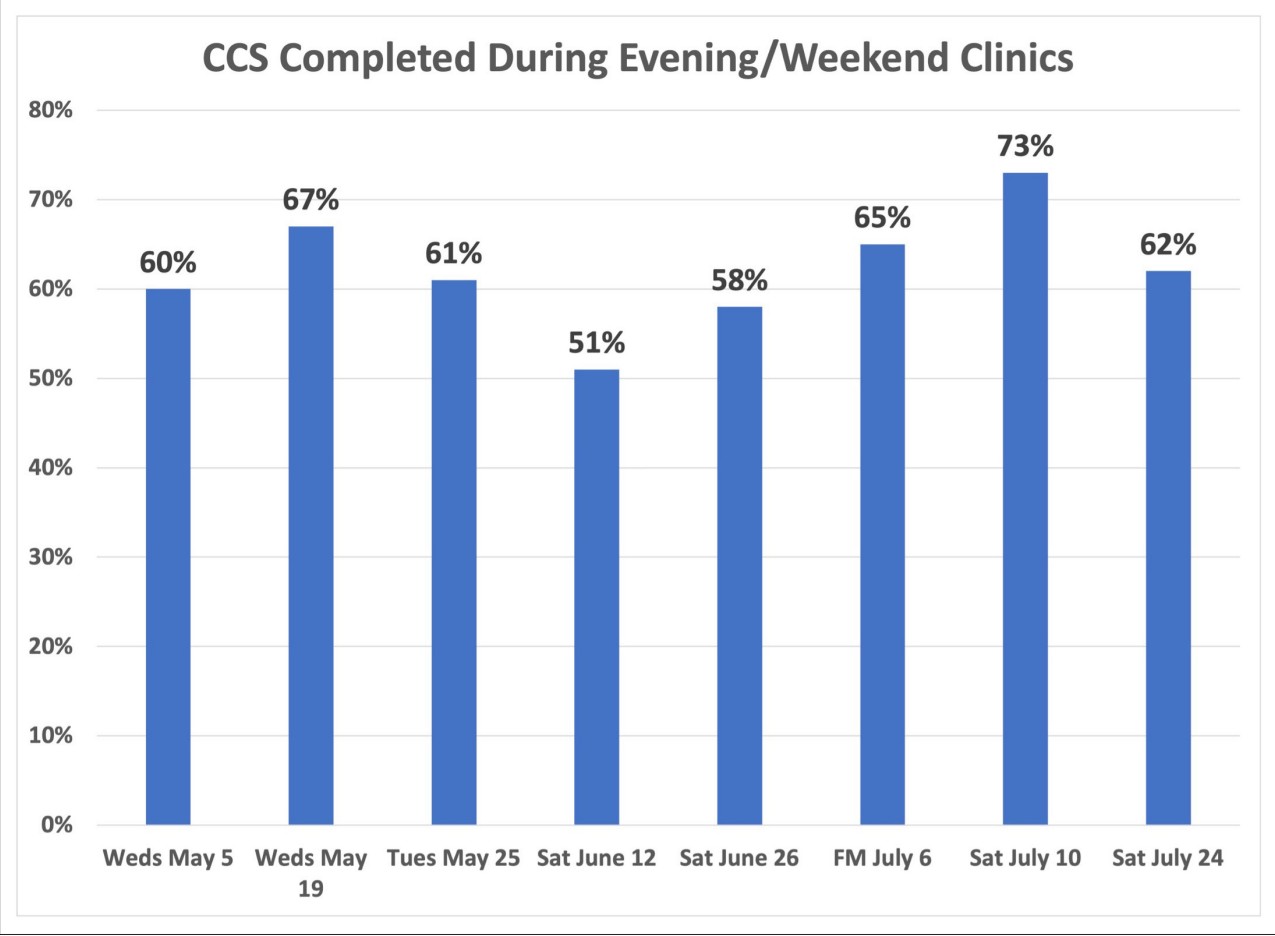

**Figure 8.** Percentage of cervical cancer screening (CCS) completed during evening/weekend clinics. CCS completion rates are reported among the patients who were scheduled into the evening/weekend clinics.

Second, we had to determine which patients needed screening. There was an estimated 15% of the overdue CCS list which was incorrect based on random chart sampling. Our chart review of over 6000 charts created a validated list to use for focused electronic outreach in the future. It also created a greater understanding for providers of how to confirm an overdue CCS and increase CCS during their clinics. Third, we had to raise awareness of the problem and of the goals of the QI project among clinic staff. Fourth, we had to develop efficient processes for completing overdue CCS. The evidence-based components chosen from the Community Guide included interventions to increase community demand (client reminders) and interventions to increase community access (extended hours) (*Centers for Disease Control and Prevention, 2019*). After establishing the validated list and presenting the project at staff meetings (provider assessment and feedback), we provided client reminders (outreach) to patients as part of normal clinic workflows or via a newly created outreach team. To increase access (extended hours), we tried the following: (1) creating new CCS clinics during regular clinic hours in the Gynecology Department; (2) creating new CCS clinics during evening/weekend hours; and (3) scheduling CCS during regular clinics in the Gynecology Department. This included patients' first scheduled appointments and rescheduled visits following missed appointments for the new CCS clinics or if patients called to schedule CCS after CCS clinics were finished. We also created workflows to communicate with AM and FM departments when patients preferred CCS to be done by the primary care provider.

We found that CCS-only clinics that focused on patients with prior abnormal results and were performed during regularly scheduled clinical hours were effective. The majority (57%) of patients who received outreach from the Gynecology Department for CCS-only clinics during regular clinic hours had their screenings done. The success of these clinics may have occurred in part because all these

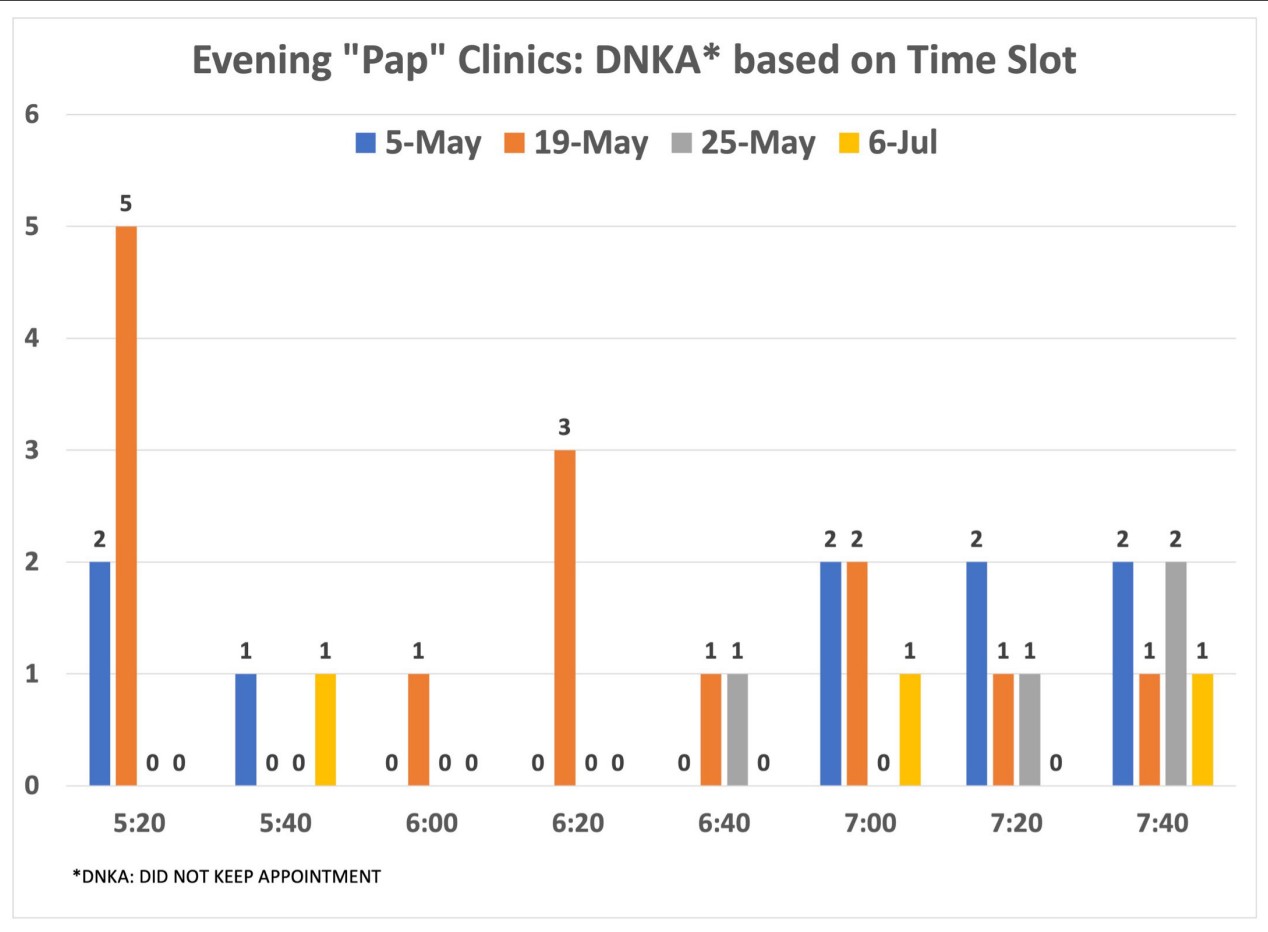

**Figure 9.** Distribution of no-show patients based on time slot: evening clinics. Number of patients who did not show up for the evening cervical cancer screening (CCS) clinics per time slot.

patients had a history of abnormal CCS and may therefore have been more knowledgeable regarding the purpose and importance of CCS. These clinics were an effective way both to increase the CCS rate and to have new providers and providers coming back from leave ease into clinical work. Resident clinics were another way to improve CCS rates while teaching pelvic exams and being financially prudent. Surveying patients was a valuable tool for obtaining patient opinions for use in improving CCS clinics and workflow. Importantly, we found that fewer than half of patients would have scheduled CCS without outreach, underscoring the importance of increasing community demand, especially in safety-net settings.

There were aspects of this project which were less effective. Most surprising was that our attempts to increase access by offering CCS appointments during evenings/weekends were not successful. Evening/weekend appointments had a much lower attendance rate (62%) compared with appointments during normal clinic hours (82%). Not all patients outreached for evening/weekend clinics had a history of abnormal CCS; therefore, some may have been less knowledgeable regarding the importance of screening. Additional reasons for low attendance may have been lack of a direct recommendation for screening from their healthcare provider, fear of pain, and low perceived need especially during a pandemic. Oyegbite et al. showed that a nurse contacting 120 patients overdue for CCS in a small practice in northwest England increased CCS rates versus texting, but the effort required to achieve this increase was unsustainable (*Oyegbite et al., 2021*). Another study of 260 patients showed a 8% increase in CCS for pregnant and postpartum patients by introducing a package of CCS information, targeted education, and widening access to screening appointments (*Coleridge et al., 2022*). Other research on CCS during the COVID-19 pandemic used different changes in workflow to improve screening rates. Martellucci et al. changed CCS appointment times from flexible scheduling

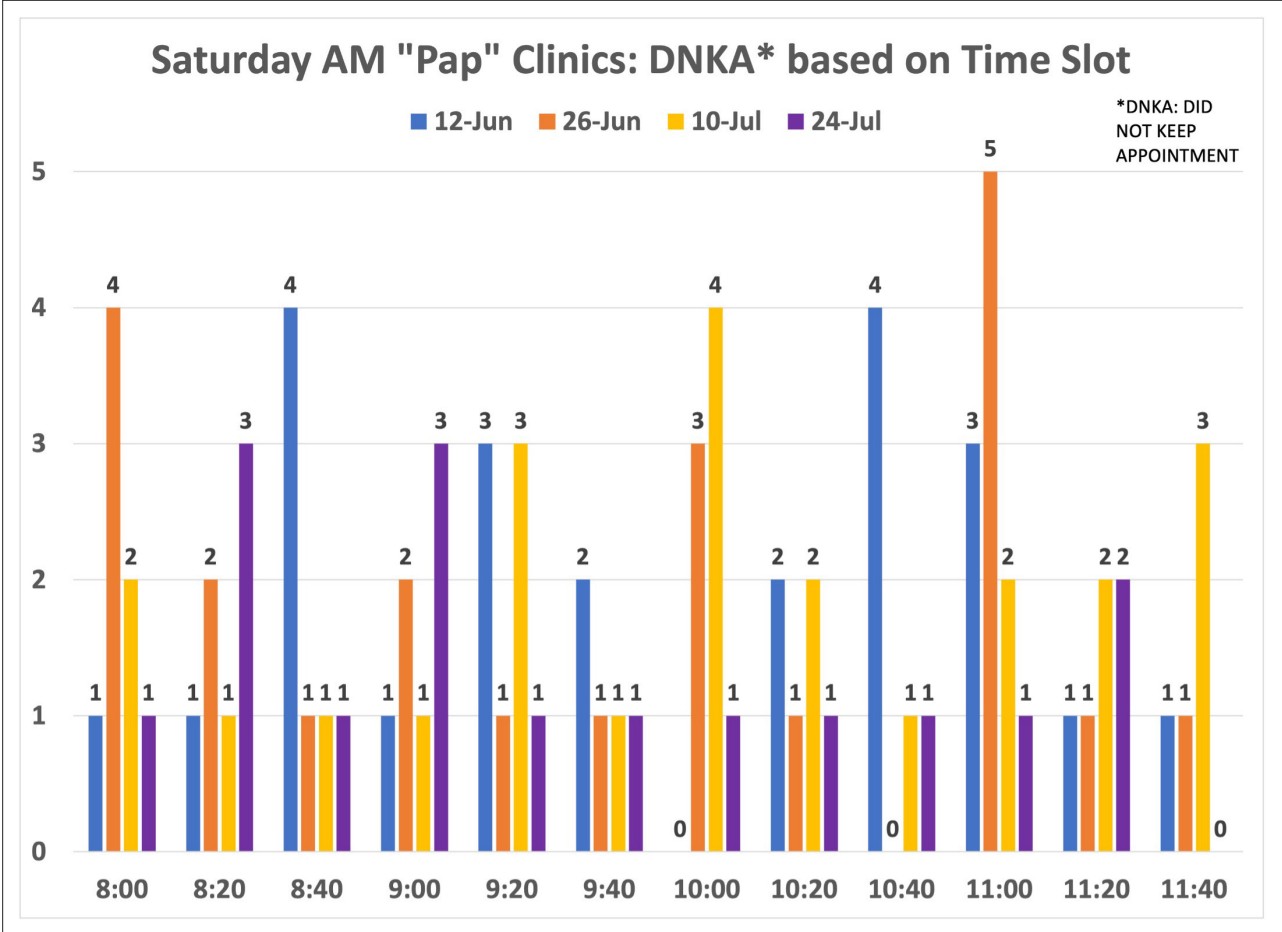

**Figure 10.** Distribution of no-show patients based on time slot: weekend clinics. Number of patients who did not show up for the weekend cervical cancer screening (CCS) clinics per time slot.

for many patients in one time slot to strict 15 min appointments for one patient only. This led to similar screening rate to pre-COVID and higher provider satisfaction (*Acuti Martellucci et al., 2021*). Castanon et al. demonstrated modeling recovery strategies for CCS emphasizing increased access and patient messaging (*Castanon et al., 2021*).

*Table 2* outlines the practical lessons gleaned from this project, which can offer meaningful guidance to other community health centers.

Future efforts to improve CCS at the EBNHC will include automated electronic targeted outreach to specific patient groups (e.g., initial screening, screened >5 years ago, and history of abnormal CCS). Optimizing the utilization of the features of an EMR system can reduce the need for work hours needed and improve the efficiency of data management and analysis. Instead of a single provider reviewing the overdue CCS EMR list, having a population health manager or grant-sponsored volunteer do the data management and analysis as a Cervical Cancer Navigator can be more efficient. Care navigation has been shown to increase cancer screening rates (*Nelson et al., 2020*). We also want to survey patients who would not schedule CCS or missed CCS appointments to create a community-specific Cervical Cancer Screening Campaign, including evaluating the role of social determinants of health. To reduce the number of patients requiring CCS each year, the EBNHC has also recently updated their CCS screening guidelines to every 3 y for 21- to 24-year-olds with cytology only and every 5 y for 25- to 65-year-olds with HPV/cytology co-testing, consistent with the American Cancer Society's 2020 guidelines (*Fontham et al., 2020*). This will allow the extension of screening intervals from 3 to 5 y for most patients. Eventual self-screening HPV testing could increase rates while optimizing resource allocation. We are currently not continuing CCS-only clinics due to the limited appointment access in all adult departments stemming from the increased need for in-person visits

**Table 2.** Lessons learned.

| |
|---|
| How many overdue CCSs does your institution have? |
| – Examine details about the numbers: # overdue CCS/provider and # overdue CCS/department to see if education is needed regarding screening guidelines or clinical workflows. |
| – Identify high-risk groups to target patient outreach: no CCS for >5 y, history of abnormal CCS, first CCS overdue. |
| |
| Why is CCS not being done? |
| – Random sampling of overdue list and deep dive into chart and identify why the CCSs were not done and change workflows accordingly. Changes in workflow should consider CCS-only clinics for new providers or providers coming back from leave. Evening/weekend CCS clinics are not necessarily the best use of resources. |
| |
| If your clinic has a high no-show/cancellation rate for CCS, why? |
| – Electronic outreach and education, language-specific messaging to these patients. |
| – Survey these patients to see why they are not coming in for CCS. |
| – Create a targeted Cervical Cancer Awareness campaign if possible. |
| |
| Educate clinical and administrative leaders |
| – Use data from your clinic's overdue CCS list to obtain buy-in from strategic stakeholders who can support changes in clinical and electronic workflows. |
| |
| Standardize workflows for rescheduling patients who decline/provider who can't get to CCS during clinic visit. |
| |
| Have a cervical cancer navigator/population health manager to oversee electronic outreach and data collection/ analysis. |
| CCS, cervical cancer screening. |

since the improvement of the COVID-19 pandemic. Restarting these clinics during regular hours in FM, AM, and the OB/GYN departments is a future aim.

This study has both strengths and limitations. Our experience working from a list of over 7000 patients is larger than similar QI projects reported in the literature. This is also one of the first successful QI projects to our knowledge specifically addressing COVID-related screening deficits in a safety net setting. However, as we describe the experience in one FQHC, our results may not be generalizable to other settings such as rural clinics or those without EMR capabilities. The EBNHC has had a provider who has been able to lead the project from the initial review of the overdue CCS EMR list to organizing the CCS clinic staffing and to create new clinical and electronic workflows to use in the future. Having dedicated staff to manage CCS and cancer screening may not be feasible for many community health centers. We were not able to include the time delay for each overdue CCS or demographic data other than age. For future studies, we would include age, race, education status, insurance status, and marital status. The cost data is proprietary/not shareable, but analysis by clinical leadership indicated the program was not cost-effective/sustainable.

## Conclusion

During the project March–August 2021, the EBNHC performed 459 CCS and increased the proportion of our total patient population who were up to date with screening by 4.5% from its nadir during 2021. The information gathered from our overdue CCS list was utilized to launch a multidisciplinary effort to learn why CCS was not being done and validate our overdue numbers. We have increased the awareness of our overdue CCS issue and regarding the EMR review needed to confirm and overdue CCS in three departments. The screenings done during the CCS project plus increased provider awareness have contributed to our increased CCS rate. We are also in the process of centralizing our

CCS workflow to decrease charting errors and make patient outreach more automated and efficient. If CCS-only clinics can be done during regular hours or resident clinics, they have value. The lessons learned from our effort can be used by other community health centers to improve CCS rates and decrease health inequities for high-risk populations in the United States.

## Additional information

### Funding
No external funding was received for this work.

### Author contributions
Sue Ghosh, Conceptualization, Data curation, Formal analysis, Investigation, Methodology, Writing - original draft, Project administration; Jackie Fantes, Supervision, Funding acquisition, Writing - review and editing; Karin Leschly, Data curation, Writing - review and editing; Julio Mazul, Conceptualization, Supervision, Methodology, Writing - review and editing; Rebecca B Perkins, Writing - review and editing

### Author ORCIDs
Sue Ghosh http://orcid.org/0000-0002-8227-3576

### Ethics
Human subjects: Reporting of aggregate data and operational details from this quality improvement project was approved by the East Boston Neighborhood Health Center Chief Medical and Chief Quality Officers.

### Decision letter and Author response
Decision letter https://doi.org/10.7554/eLife.85724.sa1
Author response https://doi.org/10.7554/eLife.85724.sa2

## Additional files

### Supplementary files
• MDAR checklist

• Source data 1. Spreadsheet data for CCS Clinics Evening/Weekends for *Figure 8* (% CCS completed/clinic), *Figure 9*: (no-show distribution of patient for CCS Evening Clinics), and *Figure 10* (no-show distribution of patient for CCS Weekend Clinics).

### Data availability
Source data for figures attached to submission.

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
