## [Editor Report]

This study addresses a pertinent and important topic related to prolonged delays in cervical cancer screening and the need to maintain routine and timely screening services in a large health maintenance network in Boston. The findings provide a solid roadmap for implementing simple strategies to help patients return to essential health services.

---

## [Decision Letter]

**Decision letter after peer review:**

Thank you for submitting your article "Electronic data review, client reminders, and expanded clinic hours for improving cervical cancer screening rates after COVID19 pandemic shutdowns: a multi-component quality improvement program" for consideration by *eLife*. Your article has been reviewed by 2 peer reviewers, and I have overseen the evaluation in my dual role of Reviewing Editor and Senior Editor.

Essential revisions:

As is customary in *eLife*, the reviewers have discussed their critiques with one another and with the Editors. The decision was reached by consensus. What follows below is the edited compilation of the essential and ancillary points provided by reviewers in their critiques and in their interaction post-review. Please submit a revised version that addresses these concerns directly. Although we expect that you will address these comments in your response letter, we also need to see the corresponding revision clearly marked in the text of the manuscript. Some of the reviewers' comments may seem to be simple queries or challenges that do not prompt revisions to the text. Please keep in mind, however, that readers may have the same perspective as the reviewers. Therefore, it is essential that you amend or expand the text to clarify the narrative accordingly.

*Reviewer #1 (Recommendations for the authors):*

The structure and flow of the manuscript could be further improved with the addition of flowcharts on the study population and the decision-making process to create a "validated" list of overdue patients.

The presentation of results and recommendations (lessons learned) is a bit confusing relative to the simplicity of data analysis and results.

*Reviewer #2 (Recommendations for the authors):*

Introduction

Pages 3-4, lines 79-83. The introduction needs more background. Can you please describe how this study fits in the literature concerning cervical cancer screening interventions? Is your study unique in that you used a combination of techniques to increase screening? You mention that the interventions you chose were from the CDC. However, some interventions may work better in certain populations or clinic sites. Why did you choose these interventions, and have they been successful in past studies?

Page 4, lines 83-85. Can you please modify the purpose to reflect what kind of study you engaged in e.g., descriptive, RCT, qualitative? It is not clear what outcome you are interested in from your current purpose statement.

Methods

The methods generally need to be edited. While this level of detail is wonderful, some of it is not necessary (e.g., I am not sure the reader needs to know staff were paid overtime).

Page 4, lines 99-100. The last statement in the first paragraph. Should this be part of the study purpose?

Page 4-5, lines 90-116. It is not clear to me how many patients ended up in your study, and these 2 paragraphs could be edited for clarity. Is it 1600? 7000? 6126? 6000? 100? (all of these numbers are mentioned). It would be helpful for an explicit statement of how many participants were included in the study.

I have no idea what kind of data you are collecting by the end of the methods. Your last paragraph should describe this clearly. Is it cervical cancer screening rates at the clinics? We need to know the kind of data you collected and how it was analyzed (i.e., did you use R software? Did you do bivariate analyses or logistic regression analyses?) How did you decide sample size?

How did you consent participants in the study?

What cervical cancer screening guidelines did you use to determine eligibility for the study? United States Preventive Services Task Force? What year? What was considered not to be up to date? e.g., 1-month or 1 year overdue?

Results

I see no description of the sample. Your population should be described (average age, race/ethnicity, etc.).

Your other results should follow what you describe as being measured in the methods which is currently not clear.

The results could benefit from editing extraneous information.

Discussion

The discussion should follow the purpose (which isn't clearly described).

---

## [Author Response]

Essential revisions:Reviewer #1 (Recommendations for the authors):The structure and flow of the manuscript could be further improved with the addition of flowcharts on the study population and the decision-making process to create a "validated" list of overdue patients.

We have added a flowchart with this information to the Results section. See Figure 2.

The presentation of results and recommendations (lessons learned) is a bit confusing relative to the simplicity of data analysis and results.

We have clarified several aspects based on Reviewer comments which we believe improves the readability of the article.

Reviewer #2 (Recommendations for the authors):IntroductionPages 3-4, lines 79-83. The introduction needs more background. Can you please describe how this study fits in the literature concerning cervical cancer screening interventions? Is your study unique in that you used a combination of techniques to increase screening? You mention that the interventions you chose were from the CDC. However, some interventions may work better in certain populations or clinic sites. Why did you choose these interventions, and have they been successful in past studies?

We have added additional reference (Han,et al., page 4, lines 95-101)and based on literature we already mentioned in the manuscript (Oyegbite and Colridge et al., page 19, lines 626-31), we focused on increasing community demand/individual outreach and increasing access by adding clinic sessions. Our focus was also based on what was feasible operationally and financially for our health institution.

Page 4, lines 83-85. Can you please modify the purpose to reflect what kind of study you engaged in e.g., descriptive, RCT, qualitative? It is not clear what outcome you are interested in from your current purpose statement.

This was a descriptive study. We have added this to the introduction (page 3, line 98-101).

MethodsThe methods generally need to be edited. While this level of detail is wonderful, some of it is not necessary (e.g., I am not sure the reader needs to know staff were paid overtime).

As the level of detail was not thought to detract from the overall paper by the editors or other reviewers, we chose to leave the description as we feel that these details may be helpful for other clinics planning to implement similar work.

Page 4, lines 99-100. The last statement in the first paragraph. Should this be part of the study purpose?

We did not feel that moving this sentence was necessary.

Page 4-5, lines 90-116. It is not clear to me how many patients ended up in your study, and these 2 paragraphs could be edited for clarity. Is it 1600? 7000? 6126? 6000? 100? (all of these numbers are mentioned). It would be helpful for an explicit statement of how many participants were included in the study.

We have added a flowchart with this information to the Results section. See Figure 2.

I have no idea what kind of data you are collecting by the end of the methods. Your last paragraph should describe this clearly. Is it cervical cancer screening rates at the clinics? We need to know the kind of data you collected and how it was analyzed (i.e., did you use R software? Did you do bivariate analyses or logistic regression analyses?) How did you decide sample size?

We have added a flowchart with this information to the Results section. See Figure 2.

How did you consent participants in the study?

This was a retrospective analysis of a QI intervention. As a QI intervention, consent was not obtained. Retrospective analysis of de-identified data was approved as noted in paper. This has been added to the methods. (page 9, line 238-42)

What cervical cancer screening guidelines did you use to determine eligibility for the study? United States Preventive Services Task Force? What year? What was considered not to be up to date? e.g., 1-month or 1 year overdue?

During the intervention, we used USPSTF 2018 guidelines. (page 4-5, lines 115-8)

ResultsI see no description of the sample. Your population should be described (average age, race/ethnicity, etc.).

As this was a QI intervention, complete demographics were not collected. We report age (page 10, line 260). We note the lack of complete demographics in the limitations (page 21-2, lines 676-8).

Your other results should follow what you describe as being measured in the methods which is currently not clear.

We have added a flowchart with this information to the Results section. See figure 2. We have added additional information as requested by reviewers.

The results could benefit from editing extraneous information.

We have not edited as it was not clear which information the Reviewer considered to be extraneous.

DiscussionThe discussion should follow the purpose (which isn't clearly described).

The purpose in the introduction (page 3, line 87-90) is mirrored in the discussion (page 16, line 556-8).